# Physical Activity Level during Pregnancy in South Africa: A Facility-Based Cross-Sectional Study

**DOI:** 10.3390/ijerph17217928

**Published:** 2020-10-29

**Authors:** Uchenna Benedine Okafor, Daniel Ter Goon

**Affiliations:** 1Department of Nursing Science, University of Fort Hare, 50 Church Street, East London, 5201, South Africa; 2Department of Public Health, University of Fort Hare, 5 Oxford Street, East London 5201, South Africa; dgoon@ufh.ac.za

**Keywords:** physical activity, pregnancy, determinants, South Africa

## Abstract

Physical activity participation during pregnancy confers many maternal and foetal health benefits to the woman and her infant and is recommended by various health bodies and institutions. However, in South Africa, scant information exists about the physical activity status and its determinants among pregnant women. The aim of this study was to assess the physical activity level and associated factors among pregnant women. This cross-sectional study enrolled 1082 pregnant women attending public health facilities in Buffalo City Municipality, Eastern Cape, South Africa. Information on socio-demographic and maternal characteristics were obtained, and the Pregnancy Physical Activity Questionnaire was used to assess physical activity during pregnancy. Multiple logistic regression analyses were used to assess associations between physical activity and the predictor variables during pregnancy. Adjusted odds ratios with 95% confidence interval were applied to estimate factors associated with physical activity levels. Statistical significance was set at *p* < 0.05. Only 278 of the women (25.7%) met recommendations for prenatal activity (≥150 min moderate intensity exercise per week). The average time spent in moderate–vigorous physical activity was 151.6 min (95% CI: 147.2–156.0). Most of the women participated in light exercises with a mean of 65.9% (95% CI: 64.8–67.0), and 47.6% (95% CI: 46.3–48.9) participated in household activities. The majority of the women did not receive physical activity advice during prenatal care sessions (64.7%). Factors negatively associated with prenatal physical activity were lower age (<19 years) (adjusted odd ratio (AOR) = 0.3; CI: 0.16–0.76), semi-urban residence (AOR = 0.8; CI: 0.55–1.03), lower educational level (AOR = 0.5; CI: 0.20–0.71), unemployment (AOR = 0.5; CI: 0.29–0.77) and nulliparity (AOR = 0.6; CI: 0.28–1.31). However, prenatal physical activity was positively associated with starting physical activity in the first trimester (AOR = 1.9; CI: 1.06–3.31) compared to other trimesters. The findings of this study demonstrated low levels of physical activity during pregnancy in South Africa. The majority of women did not meet the recommendation of 150 min of moderate intensity activity per week. Light intensity and household activities were the most preferred form of activity. The factors affecting physical activity of women in this present study include lower age, semi-urban setting, low educational level, unemployment and nulliparity. In order to increase activity levels, future work should seek to improve knowledge, access and support for physical activity in pregnant women in South Africa. This should include education and advocacy regarding physical activity for professionals involved in maternal health provision.

## 1. Introduction

The physical and psychological benefits of participation in regular physical activity during pregnancy are incontestably widely reported in the literature; they are also known to improve maternal health. As such, specialised bodies and institutions such as the American College of Obstetricians and Gynaecologists (ACOG) [1], the U.S. Department of Health and Human Services (US DHHS) [2], World Health Organization (WHO) [3], the Joint Canadian Society for Exercise Physiology (CSEP)/Society of Obstetricians and Gynaecologists of Canada (SOGC) [4], Royal Australian and New Zealand College of Obstetrics and Gynaecology (RANZCOG) [5], International Olympic Committee (IOC) [6], Department of Health & Social Care, UK [7], Sports Medicine Australia [8], and the American College of Sports Medicine (ACSM) [9] recommend and encourage women, without contraindication, to engage in moderate-intensity exercise for 150 min per week. Scientific evidence has proven the risks of physical activity during pregnancy are rare [1]. However, the rate of prenatal activity is reportedly low, both in developed [10,11,12] and developing countries [13,14,15,16,17,18]. Put differently, few women meet the authoritative guidelines of physical activity during pregnancy, perhaps, due to varying factors.

The reasons for participation in physical activity during pregnancy are multifactorial, linked to demographics, perceived risk, and obstetric care advice. For instance, studies have found older, married [13,19], lower socioeconomic status and with multiple children [16], low educational level [14], and higher income [14,20] women are less likely to be active during pregnancy. Maternal or clinical demographic variables positively associated with physical activity in pregnancy include parity [16,21], history of miscarriage [21,22], nausea [15], habitual exercise before pregnancy [23], and a higher body mass index (BMI) [24]. Despite no scientific evidence associated with adverse outcomes (e.g., preterm birth, low birth weight, miscarriage and perinatal mortality) during or following prenatal physical activity [25,26,27,28], seemingly, some pregnant women and obstetric care providers still hold doubts concerning the safety of prenatal physical activity. These findings from the literature suggest participation in physical activity during pregnancy is an interplay of multiple factors operating at different levels of the dichotomous pregnancy spectrum.

The reasons for non-participation and types of prenatal activity varied across studies and different geographical settings. Therefore, understanding the factors influencing physical activity participation in pregnancy is important to guide physical activity intervention programmes. However, unlike in other continents or regions, few data exist on prenatal physical activity in Africa, and more specifically, in South Africa. Relatedly, studies assessing the prevalence and factors associated with physical activity in South Africa have largely focused on specific subpopulations, such as children [29,30,31,32], rural populations [32,33], health professionals [34,35], and Black women [33,36], whilst few studies [37,38,39,40] are on a special population of pregnant women, with unique pregnancy-related physical, physiological, and psychosocial characteristics. Of the few studies, none utilise large samples, with heterogeneous populations, and these studies were confined to only two provinces. Notably, there have been hardly any studies undertaken to assess the level, types and intensity, and correlates of physical activity among pregnant women in the Eastern Cape Province, South Africa. This study was designed to assess the prevalence of physical activity among pregnant women attending primary health centres in Buffalo City Municipality, Eastern Cape, South Africa, and the associated factors of physical activity during pregnancy.

## 2. Methods

### 2.1. Study Design, Setting, and Participants

This was a cross-sectional descriptive study conducted among pregnant women in 12 primary health centres in Buffalo City Municipality, in the Eastern Cape Province, South Africa. Buffalo City Municipality is situated on the East Coast of the Eastern Cape Province. The details of the study setting have been described in a recent publication [41]. Briefly, the municipality is economically, one of the poorest provinces among the nine provinces in South Africa. The Buffalo City Metropolitan Municipality accounts for a total population of 884,000, or 12.2% of the total population in the Eastern Cape Province. In total, 460,000 (51.99%) of the total population are females and 424,000 (48.01%) males [42]. Buffalo City Municipality has two provincial Hospitals (Frere and Cecilia Makhiwane hospitals), and two district hospitals (Bhisho and Grey hospitals). There are five community health centres, 72 primary health clinics, and 12 mobile health services [42]. In addition, all the community health centres and primary health clinics offer antenatal healthcare services freely to all pregnant women regardless of their geographical residence, ethnic, and socio-economic background. The community and primary health facilities deliver antenatal care services every working day. Personal communication with a health facility manager revealed that, on average, the clinics register 5–6 new pregnant women who visit the primary health centres per day. Therefore, annually, an estimated 17,000 pregnant women visit the 12 selected primary health clinics for antenatal services.

### 2.2. Sample Size Determination and Sampling Procedure

We applied the Sarmah et al. [43] formulae for an infinite population to calculate the sample size at a confidence level of 95%, with the precision level of ±3%, and at a prevalence of physical activity or exercise during pregnancy of 50% (*p* = 0.5) as follows:

*p* = 0.5 and hence *q* = 1 − 0.5 = 0.5; *e* = 0.03; *z* = 1.96

So, n_0_ = $\frac{\left(1.96\right)2\;\left(0.5\right)\left(0.5\right)}{\left(0.3\right)2}=1067=1067$.

However, adding 10% non-response, the final sample size was 1215 women, to account for possible attrition and to protect against possible data loss.

We applied a two-stage sampling technique to select pregnant women, regardless of the gestation period. Firstly, using a simple random procedure, 12 antenatal primary health centres were selected to participate in the study and, secondly, participants who meet the inclusion criteria were conveniently selected because of cost and easy accessibility, since the study was conducted at the health facilities. Pregnant women were included in the study if 18 years or older, receiving antenatal care, having a single pregnancy (not multiple ones), and could read or understand the IsiXhosa, Afrikaans or English languages. Women with disabilities or reasons to cease exercise at the time of recruitment, such as “persistent excessive shortness of breath that does not resolve on rest, severe chest pain, regular and painful uterine contractions, vaginal bleeding, persistent loss of fluid from the vagina indicating rupture of the membranes, and persistent dizziness or faintness that does not resolve on rest” [44], were excluded. Detailed information about the recruitment of the participants is shown in Figure 1.

### 2.3. Ethics

The University of Fort Hare Health Research Ethics Committee approved the study protocol (Ref#2019 = 06 = 009 = OkaforUB). In addition, permission was obtained from the Eastern Cape Department of Health and all the selected health facilities. Informed consent was obtained from the pregnant women prior to data collection.

### 2.4. Data Collection

Data collection was conducted between July to October 2019. To ensure the required sample size, data collection was carried out at each antenatal health clinic on pre-specified days, in a designated room allocated to the primary researcher by the health facility manager. All eligible pregnant women attending their antenatal care visits at selected health facilities during the study period were randomly approached to participate in the study, after signing an informed consent form.

### 2.5. Main Outcome Measure: Physical Activity

The Pregnant Physical Activity Questionnaire (PPAQ) [45] was used to assess the level, type and intensity of prenatal activity. The primary outcome measure was active and inactive participation of pregnant women in physical activity during pregnancy. The PPAQ is a validated and reliable tool, widely used across countries to measure prenatal physical activity [45,46,47]. The PPAQ is comprised of 32 physical activities, which include household and care-giving (13 activities), occupational (five activities), sports and exercise (eight activities), transportation (three activities), and inactivity (three activities). We solicited participants’ participation on these different activities, and the type, intensity, duration and frequency of physical activity recorded as hours and minutes per day. To maximise the accuracy and ensure completeness of information, the PPAQ was interviewer-administered to the participants at 12 selected primary health centres during their antenatal visit and took approximately 20 to 25 min. The Metabolic Equivalent Task (MET) of each activity was categorised as sedentary (<1.5 METs), low or light (1.5– ≤3.0 METs), moderate (3.0–6.0 METs), and vigorous intensity (>6.0 METs) [45].

### 2.6. Covariates and Other Measurements

We developed a structured questionnaire to solicit information on socio-demographic, obstetrics and lifestyle behaviours of the participants. As with the PPAQ, this aspect of the questionnaire was interviewer-administered to obtain information on age, residence, ethnicity, marital status, level of education, employment status, religion, family support, and behavioural and lifestyle characteristics, which include, current exposure to alcohol and smoking. We categorised women as ‘smokers’, if they reported smoking any number of cigarettes during pregnancy, ‘non-smokers’ (reported not smoking), ‘drinkers’, as those who reported any use of alcohol during pregnancy, and ‘non-drinkers’ (reported no-drinking). Other information included whether participants had had antepartum haemorrhage in their first trimester, perceived health condition in pregnancy (women were asked how they perceive their general health: very good, good, or bad), whether participants received prenatal physical activity advice from health professionals, and had engaged in physical activity before and during pregnancy.

We obtained information on parity, mode of pregnancy delivery, and pregravid body mass index from the antenatal records of the participants.

We adopted the Institute of Medicine (IOM) recommended BMI cut-off values to classify underweight (<18.5 kg/m^2^), normal weight (18.5–24.9 kg/m^2^), overweight (25.0–29.9 kg/m^2^), and obese (>30.0 kg/m^2^) [48].

### 2.7. Data Analysis

Descriptive statistics, including mean and standard deviation (SD), median and inter-quartile range (IQR) or as proportions was applied. The Centers for Disease Control and Prevention (CDC) recommendations was used to classify women as ‘inactive’ (reporting 0–149 min of exercise per week), and ‘active’ (reported 150 min or more of physical activity) based on the combined moderate–vigorous minutes per week [49]. We applied bivariate and multivariate analyses to assess the factors affecting physical activity behaviour during pregnancy. The Chi-square was used to determine the associations between the physical activity levels and socio-demographic, lifestyle and obstetric characteristics. All the covariates associated with physical activity (that is, age, area of residence, marital status, educational level, employment status, parity, family support, mode of pregnancy delivery, antepartum haemorrhage, pre-pregnancy BMI, employment status, lifestyle behaviours, and physical activity before and during pregnancy) were included in the models. The odds ratio (OR) and corresponding confidence interval (CI) of 95% were calculated. A multiple logistic regression, using automatic variable selection procedure was applied to determine the factors that predict physical activity levels. Automatic variable selection procedures are statistical tools for choosing the best subset of predictor variables for a given response variable. The significance level was set at *p* = 0.05. The Statistical Package for Social Sciences (SPSS) (Version 24.0, IBM SPSS, Chicago, IL, USA) was used to perform all statistical analyses.

## 3. Results

### 3.1. Socio-Demographic, Obstetrics and Lifestyle Characteristics of Participants

Of the 1215 pregnant women recruited, 42 participants did not meet the eligibility criteria, 26 declined to participate, and 65 had incomplete information on medical records data so were excluded. Finally, 1082 participants were included in the analysis.

The mean age of study participants was 27.0 ± 6.2 years. The majority of the participants were aged 19–34 years (812; 75.1%), residing in an urban setting (48.3%), black (86.4%), never married (66.3%), and had attained Grade 7–12 educational (74.2%). In addition, the majority of the participants were unemployed (67.7%), Christian (89.1) and received family support (77.4%) (Table 1).

Concerning the maternal-obstetric and lifestyle characteristics, most of the participants were nulliparous (47.8%), had no antepartum haemorrhage (93.6%); and had vaginal (42.2%) and both vaginal and Caesarean section (49.7%) deliveries. The majority of the participants perceived their health as being in good condition (59.3%); an overwhelming majority had no chronic illness (93.5%), did not smoke (91.7%), nor drink alcohol (86.4%), and had normal pregravid BMI (84.8%). Of the 1082 women, 731 (67.7%) affirmed their pregnancy was not planned. The majority of the women had not received physical activity advice (700; 64.7%). In addition, 704 (65.1%) did not participate in physical activity before pregnancy, and the majority of the women (753; 69.6%) never participated in physical activity in any of the trimesters.

### 3.2. Physical Type and Intensity Levels

The women’s physical activities were compared according to type and intensity of exercise (Table 2). Descriptive analysis of physical activity scores, as derived from the PPAQ scoring regarding the level of physical activity, showed the average time spent in moderate–vigorous physical activity was 151.6 min (95% CI: 147.2–156.0). The majority of the women did not engage in moderate–vigorous intensity physical activity (i.e., sport/exercise score from PPAQ), but were physically active in light activities including household/caregiving, occupational and transportation activities.

Likewise, Table 3 presents the summary of the contribution of each physical activity level to the total score. The results show that, on average, light physical activity contributed most to the total activity. The participants rarely performed vigorous-intensity activities. Besides levels of activity, the contribution was also determined according to type of physical activity, namely, household, occupational, sport/exercise and transportation. Household activity contributed most to the total activity followed by occupational and transportation, while sport/exercise had the lowest contribution level. The rest is accounted for by physical inactivity.

### 3.3. Level of Physical Activity

Shown in Table 4 is the multivariate logistics regression analysis showing sociodemographic correlates of physical activity during pregnancy. The prevalence of moderate-intensity physical activity was 25.7% (278 out of 1082). The odds of exercise among women under 19 years of age was significantly lower than the odds of exercise among the over 34 years age group. Women aged 20–34 years were significantly less likely to exercise compared to women over 34 years age (Crude Odds Ratio (COR) = 0.1; CI: 0.07–0.21). Pregnant women who resided in semi-urban (COR = 0.5; CI: 0.41.0–0.69), were black (COR = 1.9; CI: 1.34–2.83), and married (COR = 2.5; CI: 1.63–3.81) were significantly more likely to be physically active. Unemployed women (COR = 0.3; CI: 0.26–0.45) had smaller odds of participating in physical activity during pregnancy.

The obstetric and lifestyle correlates of physical activity during pregnancy (Table 5) indicate that nulliparous women who had vaginal and caesarean section, planned pregnancy, physical activity advice, physical activity before pregnancy, and practiced physical activity during first and second trimester, were significantly more likely to be active in moderate-intensity activity during pregnancy.

In order to determine the best fitting model, an automatic variable selection procedure was used. Automatic variable selection procedures are statistical tools for choosing the best subset of predictor variables for a given response variable. The estimated logistic regression model identified age, residential area, education, employment status, parity and timing of exercise as significant predictors of physical activity during pregnancy (Table 4 and Table 5). The effects of these variables on physical activity indicated that women <19 years (AOR = 0.3; CI: 0.16–0.76), from a semi-urban area (AOR = 0.8; CI: 0.55–1.03), with primary level education (AOR = 0.5; CI: 0.20–0.71), unemployed (AOR = 0.5; CI: 0.29–0.77) and nulliparous (AOR = 0.6; CI: 0.28–1.31) were significantly less likely to participate in physical activity during pregnancy compared to older women, those from urban and rural areas, with at least a secondary school education, employed and who were primiparous or multiparous, respectively. Conversely, women who started physical activity in their first trimester (AOR = 1.9; CI: 1.06–3.31) were significantly more likely to be physically active during pregnancy compared to women commencing physical activity in the second trimester. All other variables such as race, marital status, religion, family support, previous mode of delivery, antepartum haemorrhage, smoking and consuming alcohol during pregnancy, pregravid body mass index (kg/m^2^), physical activity advice and physical activity before pregnancy were not associated with physical activity during pregnancy.

## 4. Discussion

To our knowledge, this is the first study to determine the prevalence of leisure-time physical activity participation during pregnancy and associated factors among pregnant women in the Eastern Cape Province of South Africa. Exploring the socio-demographic, behavioural, maternal and clinical factors affecting prenatal physical activity in this setting, where no previous information exist, might help in formulating policies that are key to improving maternal prenatal healthcare to this special population. From a clinical and health standpoint, physical activity participation during pregnancy confers many benefits and should be promoted, encouraged and sustained during, and even beyond pregnancy to avoid sedentary—and obesity—associated risks [16]. The interplay of socio-demographic, behavioural, maternal and clinical factors affecting physical activity during pregnancy warrant contextual understanding of the dimensionality of the factors at play to detect possible deficits and inform context-specific interventions. Contextual knowledge of factors influencing prenatal activity is a key component of the provision of quality antenatal and obstetric healthcare services for pregnant women. The findings of this study show few (25.7%) women engage in moderate-intensity physical activity. Conversely, 74.3% women do not meet the international physical activity recommendation of 150 min per week during pregnancy. The factors driving prenatal physical activity varies from one geographic setting to another.

A comparison of the rates of physical activity participation during pregnancy in this study with rates of physical activity in South Africa indicates a lower physical activity rate of 17.0% in Gauteng [39], whilst in Western province it was reported 44% and 12% women participated in light physical activity and moderate physical activity, respectively [40]. In the Watson longitudinal study of pregnant women in Gauteng province, 50.6% women were inactive in the second trimester and their physical activity decreased as pregnancy progressed [17]. Expectedly, it is thought that pregnant women in Africa and other low-income settings would exhibit relatively high physical activity, presumably because of the traditional gender-household role prescription for women, where women are to perform virtually all the household activities; however, this seems not be the case in our results. Notably, the low prenatal activity observed among women in this current study resonates with the global physical inactivity reported among pregnant women in both developed and developing countries. In addition, low prenatal physical activity have been reported in Nigeria (13.6%, 10.2%) [18,46], Ethiopia (21.9%, 8.4%) [13,21], Norway (14.6%) [50], China (11.1%) [23], and Brazil (20.1%) [16]. The lesser participation of women in moderate-intensity physical activity during pregnancy in this present study reflects the pattern of prenatal physical activity reported in low-income countries. Some studies have shown that low resources and lack of social support affect the level of physical activity practice among pregnant women in South Africa [37,40]. Clearly, the majority of the women (67.7%) in our study were unemployed, however received social support, but engage less in moderate-intensity physical activities. The issues of non-availability of facilities, resources, and lack of knowledge on physical activity and advice from healthcare providers are possible reasons for this low participation in moderate-intensity physical activity during pregnancy among women in this study. About 64.7% of the women reported receiving no physical activity advice from the healthcare providers. In addition, the wave of industrialisation and modernisation taking place in South Africa, particularly in urban areas, could serve as facilitators of physical inactivity. This partly explains the results of our finding. Since the post-apartheid era, the lifestyle behaviour of most South Africans has changed dramatically as a result of the effects of modernisation and urbanisation which promote physical inactivity and associated health risks. Generally, there is low physical activity participation among the South African populations. The South Africa National Health and Nutrition Examination Survey reported that 46% of South Africans are physically inactive [51], and pregnant women are most at high risk of being inactive and sedentary [17].

From the clinical and public health perspective, there is a need to formulate policies and programmes regarding physical activity participation during pregnancy. In the South African context, a plausible strategy to promote women’s awareness and knowledge on prenatal physical activity during pregnancy, and further encourage them to increase leisure-time physical activity, and engage in moderate-intensity physical activities is to utilise and promote the Ward-Based Outreach Teams in the community or women’s groups to promote awareness regarding the health benefits of physical activity on maternal health. This is particularly very relevant and important because the Ward-Based Outreach Teams primary focus is to promote community health by providing door-to-door health screening services and to sensitise or create awareness on various health issues in the community and schools. Community beliefs could have considerable influence on physical activity during pregnancy; therefore, efforts to enhance leisure-time, and moderate-intensity physical activity during pregnancy should be intensified. Healthcare providers (nurses, midwives, gynaecologists, physicians) should encourage women to practice moderate physical activity during pregnancy, also reinforcing the contribution of maintaining daily activities to maternal health. Pregnancy is a phase in which women are willing to alter their lifestyle behaviour, if motivated to do so.

The finding of this study shows that household activities contributed most to the total activity performed by the women. The most activities performed were of light intensity, as only a few of the pregnant women participate in moderate-to-vigorous intensity physical activity. This finding mirrors an earlier nationally representative population-based survey of 26,339 individuals, in which less than half of South Africans were moderately or vigorously physically active [52]. Our finding is consistent with other previous studies in South Africa [17,39], which reported 76.9% and 80.0% prevalence rates of household chores and walking, respectively. Similarly, household and occupational activities were reported as the most preferred physical activity performed by women during pregnancy in China [53,54], Ethiopia [13,21], Taiwan [55], Nigeria [56], Portugal [57], Serbia [14], and Brazil [16]. Pregnant women often state that family and household obligations or fear of injuries prevent them from initiating and participating in physical activity [58]; as such, it is possible that most of the pregnant women will feel more comfortable and safer doing household activities than engaging in occupational or sporting activities during pregnancy [21]. This perhaps explains the reason for the highest amount of energy being expended on household, occupational and transportation activities among pregnant women currently. In a previous study of pregnant women in South Africa, women considered physical activity as an activity of daily living, such as occupational and household tasks. They, however, perceived certain occupational, household and recreational tasks as too vigorous and consequently unsafe; the safety or otherwise of vigorous activity during pregnancy was informed by advice from family and friends, or listening to their own bodies [37]. It might be possible that the women lack access, education and support for other physical activities, which explains why majority of the women are engaged in household activities. There is need for context-specific interventions to educate and encourage pregnant women to participate in moderate-intensity physical activity to achieve optimal maternal health outcomes.

The advocacy philosophy imbibed by the UK guidelines with a dictum ‘every activity counts’ emphasising the need for inactive pregnant women to make an effort to accumulate physical activity throughout the week [59], is something to learn from and apply in the South African context. In this regard, periodic workshops on prenatal physical activity, involving a multidisciplinary team (exercise physiologists, physiotherapists, gynaecologists, obstetricians) to provide talks on the importance of engaging in, and types of moderate-intensity activities for their health and the baby is advocated. Another possible intervention strategy is to involve other exercise specialists such as exercise physiologists, physiotherapists and biokineticists (who practice exclusively in South Africa) to provide prenatal physical activity programmes, and social support within the community. By way of definition, biokineticists are exercise therapists that prescribe individualised exercise and physical activity for rehabilitation and promotion of health and quality of life [60]. In addition, utilising mHealth technology—MomConnect to promote awareness on prenatal physical activity and to encourage women to participate actively in household physical activities during pregnancy is a feasible option in South African context. The ‘MomConnect’ is a phone-based technological maternal health programme by the South African National Department of Health to support and encourage women to maintain and live healthy lifestyles during and after pregnancy, as well promote child health. The ‘MomConnect’ health promotion messages are freely available to the user. Research on exploring contextually relevant interventions to encourage pregnant women to increase leisure-time physical activity and promote moderate-intensity activities during pregnancy in this setting is desirable.

In the estimated multivariate logistic regression model, women whose age was <19 years (AOR = 0.3; CI: 0.16–0.76) were less likely to be actively engage in moderate physical activity during pregnancy compared to women 19–34 years. Conversely, in Hailemariam et al. [13] study in Ethiopia it was reported that women who were 26–35 years were 2.69 times more likely to be sedentary (AOR: 2.69, 95% CI: 1.07–6.78) compared with those in the age category of 16–25 years. Similar studies have reported age as a predictor of prenatal active practice [21,61].

Strangely, the finding of this study show that women residing in a semi-urban area (AOR = 0.8; CI: 0.55–1.03) are significantly less likely to engage in moderate physical activity compared to those who are in urban and rural settings. A recent study in Poland indicated that more pregnant women in urban than in rural areas performed physical activity during pregnancy [62]. The place of residence may have a relative influence on the extent of physical activity during pregnancy, based on safety issues, information, and infrastructural opportunities. However, these variables are out of the scope of the paper. Research exploring the factors influencing physical activity and place of residence during pregnancy would provide an insight on how best to tailor interventions to address environmental and cultural variables operating at the spectrum of prenatal activity promotion.

In this study, women with a low level of education were less likely to engage in moderate physical activity during pregnancy compared to women with a secondary level of education. This finding is similar to studies conducted in Ethiopia [13,21], Brazil [16], Serbia [14], Netherlands [63], and Nigeria [46]. In contrast, other studies found positive associations between high educational and physical activity levels [64,65,66,67]. We recognised lower levels of education in the present study. The majority of the participants have completed secondary education or have less than secondary level education (74.2%). This suggests that women with higher educational levels are better informed or have more access to knowledge about physical activity during pregnancy [21,23]. Women with lower education should be well-informed with the benefits of physical activity, and prenatal healthcare programmes should focus on prenatal physical activity to encourage women to engage in moderate-intensity physical activity for better maternal and foetal health outcomes. Providing such information to women during antenatal visits might change their attitude to prenatal physical activity and motivate them to, at least, participate in household physical activities. Women should be provided with pamphlets and possibly DVD’s that contain information on prenatal physical activities, and posters displayed at the antenatal clinics, as these will help create awareness on prenatal physical activity. We postulate that non-formal prenatal physical activity advice that may be provided to the women at the antenatal clinics in South Africa may not be sufficient, lacking in content, because not all health providers possess adequate scientific knowledge on prenatal physical activity. Therefore, adopting a systematic or formalized clinic-based prenatal physical activity intervention guide is advocated and, in this context, the training of health providers on the types, intensity, duration and physical activity recommendations during pregnancy is important, in order to bridge their knowledge gap on the subject matter.

Our findings demonstrated that being unemployed was significantly associated with a lower likelihood of prenatal physical activity compared with those who are employed. Maternal employment influences physical activity participation during pregnancy. In this study, consistent with studies conducted in Ethiopia [13], Nigeria [46] and Netherlands [63], unemployed women were more likely to be physically active than employed women. It is plausible that unemployed women have enough time to stay at home, not having the economic power to hire a domestic helper; therefore, they themselves then perform most of the household/caring activities. This is not surprising because most of the women in this present study participate in light intensity and household activities. A combination of a lower education level and household income, in this case, unemployment, entails less chance of engagement in leisure-time physical activity because the women may be lacking the social and environmental resources and information as impetus to initiate and practice physical activity [68,69]. Therefore, addressing the social and environmental factors affecting their prenatal physical activity practice would help facilitate other efforts to encourage women to participate in prenatal physical activity of moderate-intensity to improve their health and that of the unborn baby.

Our study indicated that nulliparous women are significantly less likely to be physically active during pregnancy compared to those who are primiparous or multiparous, respectively. Similar to other studies, multiparous women exhibit more physical activeness than their nulliparous peers [13,70,71], whilst previous studies have found no statistically significant association between parity and physical activity during pregnancy [72,73]. In this regard, childcare interventions are necessary to encourage women to engage in physical activity together with their children to promote the uptake of moderate-intensity physical activity during pregnancy.

However, our finding demonstrated from the predictive model that pregnant women who commence physical activity in the first trimester are significantly more likely to be physically active during pregnancy compared to those who start later than the second trimester. This is concordant with a study in Poland [61], and USA [15] which found the odds of meeting physical activity recommendations, including both moderate and vigorous activities, to be higher among women in the first trimester compared to the third trimester. However, our finding is inconsistent with the findings from other previous studies, which indicated that women in the third trimester were more likely to meet the physical activity guideline compared to those in the first trimester [17,54,65,74,75]. Although the present study did not probe for the cessation of physical activity, notably, the decrease in physical activity in the second and the third trimesters of pregnancy can be attributed to women’s mood changes and the unborn baby’s foetal growth, which result in weight gain and discomforts such as fatigue, back pain, and sleeplessness [76,77]. Women should be encouraged to engage and sustain their prenatal activity throughout pregnancy, and beyond. This is particularly important and relevant to women with no pregnancy complications and contraindications; pregnant women, therefore, should be advised to undertake moderate-intensity physical activity, accumulating at least 150 min per week, as stipulated by international bodies.

## 5. Implications

Our findings have emphasised the need to tailor health promotion interventions in the promotion of moderate-intensity physical activity in the antenatal healthcare as part of the primary healthcare agenda. Healthcare providers should regard physical activity as a prescription, rather than an option to be observed. The findings highlight several factors associated with low level of prenatal physical activity among pregnant women in this setting, which call for a context-specific and multidisciplinary approach to the promotion of prenatal activity, particularly focusing on household and moderate-intensity activities. Understandably, the demographic correlates of prenatal physical activity provide useful information on the variables affecting prenatal physical activity; such information is relevant to inform interventions and to identify those who need intervention. For example, as in this case, interventions focusing on women with a low level of education and without employment are imperative. Stakeholders, such as the Eastern Cape Department of Health, research organizations, local NGOs working with pregnant women, and faith-based organisations or societies should provide relevant and appropriate strategies that include educational interventions to address barriers to moderate physical activity during pregnancy. Notably, programs or pamphlets, posters showing information and instructions on physical activity during pregnancy are absent in the routine of the pregnant population in South Africa. As such, it is desirable that health policy makers, health providers, and stakeholders working with pregnant population prioritise physical activity for all pregnant women.

## 6. Limitations

One obvious limitation of the study is the use of a self-reported questionnaire, which could have inherent bias in the manner of responses, and the PPAQ questionnaire was not validated locally against objective methods; as such, we cannot draw conclusion on its measurement properties (reliability, criterion validity, construct validity, responsiveness) in the South African context. Therefore, future studies on the validity of the PPAQ in this population are warranted, and studies using objective measure, such as an accelerometer to evaluate physical activity during pregnancy. Furthermore, the PPAQ is reported to have insufficient construct validity in assessing total and vigorous physical activity scores in pregnancy, however, with low-to-moderate quality evidence [78]. In addition, the participants were pregnant women attending public health facilities in Buffalo City Municipality of the Eastern Cape; as such, this limited the generalisability of the findings to the entire Eastern Cape Province. Despite these limitations, our study provides useful information for future comparative studies on the factors, types and intensity of physical activity participation during pregnancy in the Eastern Cape Province of South Africa. We believe such information would be relevant in shaping maternal health interventions in the context of promotion of leisure-time physical activity, at least in this setting. Again, the prospective evaluation of physical activity using the PPAQ, a simple and short, reliable and valid tool, which has been used in several countries [47,79,80,81,82,83,84], and the large sample of pregnant women serve as unique strengths of the study. To the best of the author’s knowledge, previous studies examining physical activity during pregnancy in South Africa have not utilised such a large population as ours. In addition, contrary to other previous published studies, our study involved all racial groups in the South African context with varying sociodemographic characteristics.

## 7. Conclusions

The findings of this study demonstrated low participation of women in moderate-intensity physical activity during pregnancy. The majority of the women did not meet the 150 min per week recommendations. Light intensity and household activities were the predominant forms of physical activity among pregnant women. The factors affecting physical activity of women in this present study included age, residential area, lower educational level, unemployment, nulliparity and trimester timing of exercise as significant predictors of physical activity during pregnancy. There exists a need to provide basic education to mothers, and a general advocacy campaign on the health benefits of moderate-intensity physical activity during pregnancy for optimal participation in prenatal activity should also be launched.

## Figures and Tables

**Figure 1 ijerph-17-07928-f001:**
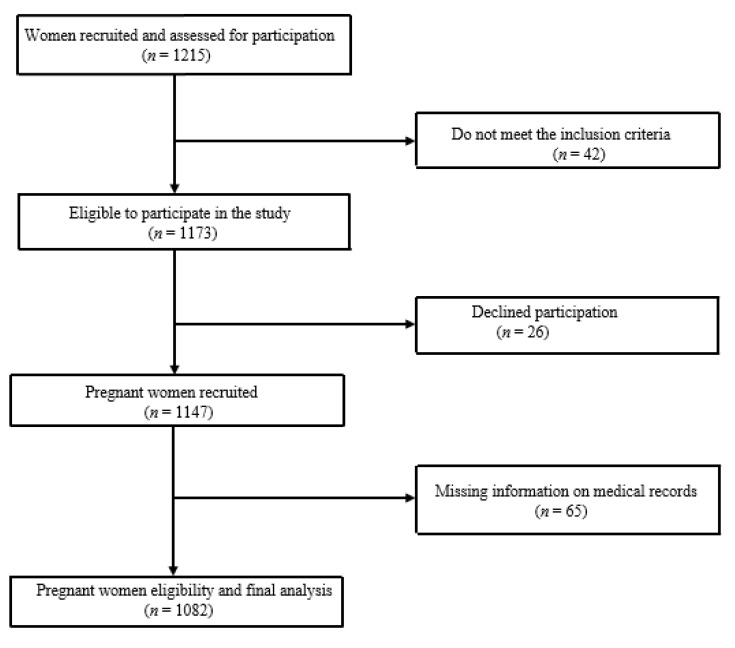
Flow diagram of sample selection and participation.

**Table 1 ijerph-17-07928-t001:** Socio-demographic, obstetrics and lifestyle characteristics of participants.

Variables	Frequency	Percentage
Age (years)		
<19	118	10.9
19–34	812	75.1
>34	152	14.0
Residential area		
Rural	118	10.9
Semi-urban	441	40.8
Urban	523	48.3
Race		
Black	935	86.4
Coloured	147	13.6
Marital status		
Married	236	21.8
Never married	717	66.3
Cohabiting	129	11.9
Educational level		
Primary	84	7.8
Secondary	803	74.2
Tertiary	195	18.0
Employment status		
Unemployed	733	67.7
Employed	349	32.3
Religion		
Christian	964	89.1
Other	118	10.9
Family support		
Adequate	837	77.4
Moderate	223	20.6
No support	22	2.0
Parity		
Nulliparous	517	47.8
Primiparous	322	29.8
Multiparous	243	22.4
Delivery mode		
Vaginal	457	42.2
Caesarean section	87	8.1
Both	538	49.7
Antepartum haemorrhage		
Yes	69	6.4
No	1013	93.6
Perceived health condition		
Very Good	440	40.7
Good	631	58.3
Bad	11	1.0
Smoking status		
Yes	90	8.3
No	992	91.7
Alcohol use		
Yes	147	13.6
No	935	86.4
Pregravid body mass index (kg/m^2^)		
Underweight (<18.5)	22	2.0
Normal weight (18.5–24.9)	917	84.8
Overweight (25.0–29.9)	143	13.2
Planned pregnancy		
Yes	351	32.4
No	731	67.6
Physical activity advice		
Yes	382	35.3
No	700	64.7
Physical activity before pregnancy		
Yes	378	34.9
No	704	65.1
Physical activity timing		
1st trimester	188	17.4
2nd trimester	114	10.5
3rd trimester	27	2.5
Never	753	69.6

**Table 2 ijerph-17-07928-t002:** Physical activity levels during pregnancy (minutes).

Activity Category	Mean	LCL	UCL	Minimum	Quartile 1	Median	Quartile 3	Maximum
Total energy expenditure	151.6	147.2	156.0	22.3	100.0	138.9	191.1	546.2
Total light	128.7	124.3	133.1	8.2	74.2	114.0	166.3	541.8
Subdivision by intensity								
Sedentary	22.9	21.9	24.0	0.0	7.4	18.9	31.9	88.2
Light	99.6	96.5	102.7	7.1	61.3	89.6	131.8	314.5
Moderate	28.3	25.9	30.7	0.0	2.5	12.2	37.6	378.0
Vigorous	0.8	0.6	1.0	0.0	0.0	0.0	0.8	40.5
Subdivision by type of activity								
Household/caregiving	73.8	70.6	77.0	0.0	35.0	65.3	92.8	413.0
Occupational	25.1	23.2	27.1	0.0	1.8	10.5	44.8	308.0
Sport/exercise	2.6	2.3	2.9	0.0	0.0	0.8	3.2	65.7
Transportation	23.1	21.5	24.6	0.0	3.4	14.0	35.0	157.5
Inactivity	27.0	25.8	28.2	0.0	14.0	18.9	41.5	98.0

LCL = Lower confidence level; UCL = Upper confidence level.

**Table 3 ijerph-17-07928-t003:** Contribution of each type activity to the total activity score.

Activity Category	Mean	LCL	UCL	Minimum	Quartile 1	Median	Quartile 3	Maximum
Subdivision by intensity								
Light	65.9	64.8	67.0	8.5	53.1	65.9	80.2	100.0
Moderate	15.8	14.8	16.7	0.0	1.6	10.9	27.0	74.2
Vigorous	0.5	0.4	0.6	0.0	0.0	0.0	0.4	19.9
Subdivision by type of activity								
Household/caregiving	47.6	46.3	48.9	0.0	31.0	47.3	63.7	97.7
Occupational	16.0	15.0	17.0	0.0	1.2	9.2	28.5	78.7
Sport/exercise	1.7	1.5	1.9	0.0	0.0	0.6	2.1	31.2
Transportation	14.0	13.2	14.8	0.0	3.3	10.2	20.9	77.9
Inactivity	20.8	19.8	21.7	0.0	8.0	17.4	30.5	98.0

LCL = Lower confidence level; UCL = Upper confidence level.

**Table 4 ijerph-17-07928-t004:** Multivariate logistics regression model of physical activity on sociodemographic characteristics.

Variables	Total (*n* = 1082)	Active (*n* = 278)	Inactive (*n* = 804)	COR (95% CI)	AOR (95% CI)
	*n* (%)	*n* (%)	*n* (%)		
Age (years)					
<19	118 (10.9)	14 (11.9)	104 (88.1)	0.1 (0.07–0.21) *	0.3 (0.16–0.76) *
19–34	812 (75.1)	168 (20.7)	644 (79.3)	0.3 (0.22–0.42) *	0.9 (0.57–1.40)
>34	152 (14.0)	96 (63.2)	56 (36.8)	1	
Residential area					
Rural	118 (10.9)	60 (50.8)	58 (49.2)	1.4 (0.97–1.88)	1.2 (0.78–2.00)
Semi-urban	441 (40.8)	111 (25.2)	330 (74.8)	0.5 (0.41–0.69) *	0.8 (0.55–1.03) *
Urban	523 (48.3)	107 (20.5)	416 (79.5)	1	
Race					
Black	935 (86.4)	233 (24.9)	702 (75.1)	1.9 (1.34–2.83) *	1.2 (0.71–2.12)
Other	147 (13.6)	45 (30.6)	102 (69.4)	1	
Marital status					
Married	236 (21.8)	75 (31.8)	161 (68.2)	2.5 (1.63–3.81) *	1.4 (0.84–2.49)
Never married	717 (66.3)	160 (22.3)	557 (77.7)	1.0 (0.74–1.38)	1.3 (0.79–2.12)
Cohabiting	129 (11.9)	43 (33.3)	86 (66.7)	1	
Educational level					
Primary	84 (7.8)	29 (34.5)	55 (65.5)	0.5 (0.28–0.72) *	0.4 (0.20–0.71) *
Secondary	803 (74.2)	181 (22.5)	622 (77.5)	0.4 (0.31–0.57) *	0.7 (0.45–0.95) *
Tertiary	195 (18.0)	68 (34.9)	127 (65.1)	1	
Unemployment status					
Unemployed	733 (67.7)	171 (23.3)	562 (76.7)	0.3 (0.26–0.45) *	0.5 (0.29–0.77) *
Employed	349 (32.3)	107 (30.7)	242 (69.3)	1	
Religion					
Christian	964 (89.1)	234 (24.3)	730 (75.7)	1.4 (0.93–2.04)	0.9 (0.53–1.39)
Other	118 (10.9)	44 (37.3)	74 (62.7)	1	
Family support					
Adequate	837 (77.4)	212 (25.3)	625 (74.7)	0.4 (0.28–0.59) *	1.0 (0.37–2.71)
Moderate	223 (20.6)	57 (25.6)	166 (74.4)	0.2 (0.09–0.28) *	0.7 (0.25–1.92)
No support	22 (2.0)	9 (40.9)	13 (59.1)	1	

COR = Crude Odds Ratio, AOR = Adjusted Odds Ratio, CI = Confidence Interval; * Statistically significant.

**Table 5 ijerph-17-07928-t005:** Multivariate logistics regression model of physical activity on obstetric and lifestyle characteristics.

Variables	Total (*n* = 1082)	Active (*n* = 278)	Inactive (*n* = 804)	COR (95% CI)	AOR (95% CI)
	*n* (%)	*n (*%)	*n (*%)		
Parity					
Nulliparous	517 (47.8)	98 (19.0)	419 (81.0)	0.3 (0.24–0.44) *	0.6 (0.28–1.31) *
Primiparous	322 (29.8)	102 (37.3)	220 (68.3)	0.8 (0.54–1.06)	0.7 (0.45–1.03)
Multiparous	243 (22.4)	78 (32.1)	165 (67.9)	1	
Previous mode of delivery					
Vaginal	457 (42.2)	127 (27.8)	330 (72.2)	2.4 (1.89–3.16) *	1.7 (0.86–3.54)
Caesarean section	87 (8.1)	52 (59.8)	35 (40.2)	3.3 (2.31–4.69) *	1.7 (0.72–3.82)
Both	538 (49.7)	99 (18.4)	439 (81.6)	1	
Antepartum haemorrhage					
Yes	69 (6.4)	33 (47.8)	36 (52.2)	1.2 (0.72–1.91)	1.2 (0.66–2.05)
No	1013 (93.6)	245 (24.2)	768 (75.8)	1	
Perceived health condition					
Very Good	440 (40.7)	87 (19.8)	353 (80.2)	0.3 (0.21–0.48) *	0.2 (0.05–0.91)
Good	631 (58.3)	185 (29.3)	446 (70.7)	0.6 (0.39–0.81) *	0.4 (0.09–1.63)
Bad	11 (1.0)	6 (54.6)	5 (45.4)	1	
Chronic illness					
Yes	70 (6.5)	35 (50.0)	35 (50.0)	1.3 (0.79–2.09)	1.0 (0.54–1.68)
No	1012 (93.5)	243 (24.0)	769 (76.0)	1	
Smoking status					
Yes	90 (8.3)	26 (28.9)	64 (71.1)	0.5 (0.3–0.79) *	1.1 (0.55–2.14)
No	992 (91.7)	252 (25.4)	740 (74.6)	1	
Alcohol use					
Yes	147 (13.6)	48 (32.6)	99 (67.4)	0.6 (0.39–0.82) *	0.8 (0.53–1.27)
No	935 (86.4)	230 (24.6)	705 (75.4)	1	
Pregravid body mass index (kg/m^2^)					
Underweight (<18.5)	22 (2.0)	4 (18.2)	18 (81.8)	0.1 (0.05–0.35) *	0.2 (1.07–1.11)
Normal weight (18.5–24.9)	917 (84.8)	200 (21.8)	717 (78.2)	0.9 (0.62–1.14)	0.9 (0.69–1.34)
Overweight (25.0–29.9)	143 (13.2)	74 (51.7)	69 (48.3)	1	
Planned pregnancy					
Yes	351 (32.4)	101 (28.8)	250 (71.2)	1.9 (1.48–2.47) *	1.2 (0.81–1.64)
No	731 (67.6)	177 (24.2)	554 (75.8)		
Physical activity advice					
Yes	382 (35.3)	116 (30.4)	266 (69.6)	2.1 (1.66–2.76) *	1.3 (0.92–1.72)
No	700 (64.7)	162 (23.1)	538 (76.9)	1	
Physical activity before pregnancy					
Yes	378 (34.9)	123 (32.5)	255 (67.5)	2.7 (2.1–3.51) *	1.3 (0.77–2.05)
No	704 (65.1)	155 (22.0)	549 (78.0)	1	
Physical activity gestational period					
1st trimester	188 (17.4)	62 (33.0)	126 (67.0)	3.2 (2.34–4.47) *	1.9 (1.06–3.31) *
2nd trimester	114 (10.5)	45 (39.5)	69 (60.5)	3.0 (2.03–4.37) *	2.0 (1.10–3.70) *
3rd trimester	27 (10.5)	15 (55.6)	12 (44.4)	1.3 (0.79–2.07)	1.0 (0.38–2.57)
Never	753 (69.6)	156 (20.7)	597 (79.3)	1	

COR = Crude Odds Ratio, AOR = Adjusted Odds Ratio, CI = Confidence Interval; * Statistically significant.

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
