# Peer review of "Physical Activity Level during Pregnancy in South Africa: A Facility-Based Cross-Sectional Study"

_ijerph, 2020, doi:10.3390/ijerph17217928_

Round 1
Reviewer 1 Report
This article is very interesting. Currently it is necessary to promote healthy habits during pregnancy, among which is the practice of physical exercise, which favors the creation of healthy habits, which are extended to the whole family.
The results of this study provide valuable information when evaluating the level of PA in South African pregnant women and its associated factors.
The article is well founded, with a well-defined theoretical framework and addresses a current problem in society, the importance of physical exercise in pregnant women.
Its results are important, although it is not considered that they could be extrapolated to the entire population of pregnant women. But it adequately analyzes the absence of physical exercise in this population group.
The article is well written and following the appropriate regulations, although it presents an important question that the authors must resolve. The categorization of BMI according to the IOM is not correctly expressed, a fact that may affect the results obtained.
The design of the article is correct, although it should be clear if all the women who attended the health centers were chosen and if not, what was the method of randomization of the women, and express it in the text. It would have been necessary for the authors to express the week of gestation in which the women were surveyed
On line 23, the authors write that the mean BP was 151.6 minutes. If only 25.7% of the women met the recommendations for physical exercise during pregnancy, how is it that the average is 151.6 minutes?
In figure 1, and line 196, it would be convenient to know the reason why the 26 women refused to participate, which would support the theoretical framework of the article.
On line 171, one of the variables that is collected is the perceived state of health during pregnancy. It would be convenient to state whether this state of health was collected by means of a validated questionnaire, and the parameters that are collected in the table should be clarified as the woman comes to consider her state of health.
In line 174, the authors adopt the cutoff values ​​for BMI recommended by the Institute of Medicine (IOM), it is convenient that the authors review the values ​​recommended by the IOM and correct those expressed in the article.
The data expressed in table 1 referring to BMI do not reflect those expressed in line 174, we ask that they be corrected once the previous point has been reviewed.
Author Response
Reviewer #1
On behalf of my co-author, I would like to express appreciation for the thorough review and editorial work conducted on our manuscript. The comments provided by you are worthwhile; and has added significantly to the improvement of the manuscript. We have tried to address the queries point by point as instructed and made appropriate amendments as necessary.
Once again, many thanks.
General comments
This article is very interesting. Currently it is necessary to promote healthy habits during pregnancy, among which is the practice of physical exercise, which favors the creation of healthy habits, which are extended to the whole family.
The results of this study provide valuable information when evaluating the level of PA in South African pregnant women and its associated factors.
The article is well founded, with a well-defined theoretical framework and addresses a current problem in society, the importance of physical exercise in pregnant women.
Its results are important, although it is not considered that they could be extrapolated to the entire population of pregnant women. But it adequately analyzes the absence of physical exercise in this population group.
Response
The above comments are greatly appreciated.
Comment
The article is well written and following the appropriate regulations, although it presents an important question that the authors must resolve. The categorization of BMI according to the IOM is not correctly expressed, a fact that may affect the results obtained.
Response
Thank you for this keen observation. This was an oversight. This has been revised and also corrected in Tables 1 and 5 accordingly.
We adopted the Institute of Medicine (IOM) recommended BMI cut-off values to classify underweight (18.5kg/m2), normal weight (18.5-24.9kg/m2), overweight (25.0-29.9kg.m2), and obese (>30.0 kg/m2) [45].
Comment
The design of the article is correct, although it should be clear if all the women who attended the health centers were chosen and if not, what was the method of randomization of the women, and express it in the text. It would have been necessary for the authors to express the week of gestation in which the women were surveyed.
Response
All pregnant women attending antenatal health care in the selected health centers who met the eligibility criteria approached for participation in the study. This we have stated it in the text as follows: “We applied a two-stage sampling technique to select pregnant women. Firstly, using simple random procedure, 12 antenatal primary health centres were selected to participate in the study; and secondly, participants who meet the inclusion criteria were conveniently selected because of cost and easy accessibility, since the study was conducted at the health facilities.” We stated in the data collection section that “All eligible pregnant women attending their antenatal care visits at selected health facilities during the study period were randomly approached to participate in the study, after signing an informed consent form”.
As regard to the gestational period, all the three trimesters of pregnancy were included. As per the suggestion above, we have added it in the text as follows:
We applied a two-stage sampling technique to select pregnant women, regardless of the gestation period.
Comment
On line 23, the authors write that the mean BP was 151.6 minutes. If only 25.7% of the women met the recommendations for physical exercise during pregnancy, how is it that the average is 151.6 minutes?
Response
The percentage of 25.7% (n=278) of the women meeting the recommendations was calculated using the Centers for Disease Control and Prevention (CDC) recommendations which is used “to classify women as ‘inactive’ (reporting 0-149 minutes of exercise per week), and ‘active’ (reported 150 minutes or more of PA) [46]” However, the majority of the women did not engage in moderate physical activity (i.e. sport/exercise score from PPAQ), but were physically active in light activities including household/caregiving, occupational and transportation activities.
Comment
In figure 1, and line 196, it would be convenient to know the reason why the 26 women refused to participate, which would support the theoretical framework of the article.
Response
Based on ethical principle of conducting research, we did not probed the participants to ascertain reasons for their refusal to participate in the research.
Comment
On line 171, one of the variables that is collected is the perceived state of health during pregnancy. It would be convenient to state whether this state of health was collected by means of a validated questionnaire, and the parameters that are collected in the table should be clarified as the woman comes to consider her state of health.
Response
The perceived health state of health during pregnancy as one of the demographic variable was included to ascertain the general health condition of the women regarding how they perceived their health condition as ‘very good’, ‘good’ or ‘bad’. Since information on the status of the pregnancy, such as nausea and recommendations for bedrest were not collected, the perceived health condition serve as a proxy. The narrative is as follows:
We developed a structured questionnaire to solicit information on socio-demographic, obstetrics and lifestyle behaviours of the participants. As with the PPAQ, this aspect of the questionnaire was interviewer-administered to obtain information on age, residence, ethnicity, marital status, level of education, employment status, religion, family support; and behavioural and lifestyle characteristics, which include, current exposure to alcohol and smoking. We categorised women as ‘smokers’, if they reported smoking any number of cigarettes during pregnancy, ‘non-smokers’ (reported not smoking), ‘drinkers’, as those who reported any use of alcohol during pregnancy, and ‘non-drinkers’ (reported no-drinking). Other information included whether participants had had antepartum haemorrhage in their first trimester, perceived health condition in pregnancy (women were asked how they perceive their general health: very good, good, or bad), whether participants received prenatal physical activity advice from health professionals, and had engaged in physical activity before and during pregnancy.
Comment
In line 174, the authors adopt the cutoff values ​​for BMI recommended by the Institute of Medicine (IOM), it is convenient that the authors review the values ​​recommended by the IOM and correct those expressed in the article.
Response
Thank you for the observation.
The cutoff values ​​for BMI recommended by the Institute of Medicine (IOM) has been revised as thus
|
Underweight (<18.5) |
|
Normal weight (18.5-24.9) |
|
Overweight (25.0-29.9) |
The incongruence in the text and the table has been harmonised accordingly. In addition, the pre-pregnancy BMI has been recalculated.
Comment
The data expressed in table 1 referring to BMI do not reflect those expressed in line 174, we ask that they be corrected once the previous point has been reviewed.
Response
Thank you for the observation. This inconsistence has been corrected as indicated above.

Reviewer 2 Report
The authors present findings from a quantitative assessment of prenatal physical activity behavior in a large cohort of pregnant individuals residing in the Eastern Cape, South Africa. The study does add to the scientific literature base, but really only to support that activity levels remain low in this population (as shown in nearly all pregnant populations around the world). The authors do complete a multivariate analysis to identify the demographic factors behind low activity levels in this region, and provide some ideas on interventions on how to improve physical activity during pregnancy. The manuscript is interesting but needs considerable revision to improve overall precision, clear rationale/explanations, and implications from this work.
At present, the introduction, results and discussion require significant editing for readability.
General comments:
- Consider avoiding acronym ‘PA.’ General consensus is that acronyms should be avoided where possible and if not established. PA won’t necessarily be recognised as ‘physical activity’ to researchers outside of this field.
- Instead of terming population as ‘pregnant women,’ authors should consider using more inclusive language in their manuscript (unless of course the authors asked all participants about their gender identity and they all responded ‘woman,’ if so, should be mentioned in the methods/results).
- g. Physical Activity Level during Pregnancy in South Africa: A Facility-Based Cross-Sectional Study
- Opening line of abstract: Physical activity (PA) participation during pregnancy confers many maternal and fetal health benefits.
Specific comments:
Abstract
- Consider avoiding acronym ‘PA.’ It may add to word count of abstract, but there are statements that could be removed to aid this change e.g. Line 13 – ‘as most studies have focused on the general population.’ Information not really needed.
- Line 20, ‘Statistical significance was set at p< 0.05 level.’ The word ‘level’ not needed.
- Line 21: what is the international guideline for prenatal activity? Do mean the general consensus of guidelines from around the world e.g. ACOG, CSEP, IOC, or do you mean the IOC? Either be specific as to which guideline or reword to something like ‘met recommendations’ for a more generic statement.
- Line 22, ‘the majority of women did not receive PA advice’ would be better following the statements regarding achievement of recommendations/actual activity levels (i.e. after sentence ending ‘…household activities..’
- Line 19 – acronym needed for adjusted odds ratio – AOR used later in abstract.
- Line 30 – low levels of PA during pregnancy in South Africa. Add to be specific, it is already pretty well established that PA is low in pregnancy, but this study is adding to that literature by focusing on a geographic location.
- Lines 31 to 34 is very repetitive of the results. I would consider condensing these together and using the final sentences to focus more on implications. I would like to see expansion on ideas of how to improve PA in this population.
Introduction
- Lines 42 to 44 – you should also include references to the IOC recommendations, CSEP/SOGC, UK guidelines, etc. if you want to the guidelines around the world appropriately.
- Para 2 Lines 50 to 68 – This is all interesting and relevant information, but it could be considerably condensed. I think the authors just need to state reasons for PA participation are multifactorial - linked to demographics, perceived risk, and obstetric care advice, with some clear examples without extensive literature review.
- Para 3 Lines 80 – 87 – Again, interesting information but not necessarily needed for the introduction – I think this could be moved to the discussion if not already included. It detracts from your rationale.
Methods
- Line 94 ‘Buffalo City…’ direct repetition of line 93. Remove.
- Line 125 – contraindications list. This is incorrect - these are not contraindications to prenatal exercise – these are reasons to cease activity if exercising. Contraindications include health conditions like preeclampsia, growth restriction, etc. Please check the reference again and amend. The ACOG, CSEP/SOGC and IOC have a clear list of contraindications to prenatal exercise. If you did not one of these lists to act as exclusion, then you must state that you did not exclude by contraindications, but reasons to cease exercise.
- Figure 1 – what is the difference between incomplete PPAQ and missing data? I think you could benefit from stating what data was missing – was it antenatal outcome records? Also, if the PPAQ was interviewer-administered, how was it incomplete? I think this needs to be explained as your methods specifically states on Line 155-156 ‘To maximise the accuracy and ensure completeness of information, the PPAQ was interviewer-administered…’
- Line 175 - BMI underweight category is incorrect.
Results
- Line 207 to 209 – I think the statements about unplanned pregnancy and then PA advice, PA participation should be separated. They aren’t necessarily related and shouldn’t be in the same sentence.
- Table 1 – Would move headings to left-aligned. It is hard to read centralized with groupings beneath. This may be formatting edited later.
- Table 4 and results section 3.3 – there is a lot of repetition for reporting of odds ratios in the text and in the table. It is not needed. The odds ratio should only be written in the text if they are not presented in the table or a figure. This is also applicable to the other results sections, but it is particularly noticeable here. It would make the results more succinct and far easier to read.
- Furthermore, more focus should be placed on the adjusted odds ratios than the crude odds ratios, at the moment the reporting of the crude odds ratios is overwhelming.
- I am not quite understanding how 704 (65.1%) did not participate in PA during pregnancy, and the majority of the women (753; 69.6%) never participated in PA in any of the trimesters but the mean PA was 151 minutes per week (above recommendations!) and the median was 138.9. This doesn’t really fit with the ‘story.’ I think it is down to the wording. I would suggest being more precise with what PA is being referred too throughout the manuscript e.g. The majority of the population did not engage in moderate physical activity (i.e. sport/exercise score from PPAQ), but were physically active in light activities including household/caregiving, occupational and transportation activities. That is the distinction needed. The population are not meeting ‘exercise’ recommendations.
- On this note, more focus should be placed on encouraging pregnant individuals to increase leisure-time physical activity, and engaging in moderate activities.
Discussion
- Specific focus on rephrasing to specify ‘moderate-intensity’ or ‘leisure time’ physical activity throughout.
- The discussion is far too long and repetitive. There is a lot of information that provides insight but actually is not additive to the interpretation of these findings in this context. It feels more like an extended literature review than a discussion. Additionally, I often find the authors jump between different ideas within a paragraph with no clear link and this makes it hard to read. This section could be improved by condensing the ideas considerably and providing discussion of the most relevant works related to this study. Paragraphs 2, 4 and 5 need particular work.
- Line 307 – ‘This study used..’ comes out of nowhere, should be removed and addressed just in the limitations.
- There are a lot of emotive words e.g. worryingly, disappointingly – I think these should be removed. How physical activity messaging and engagement is shared with pregnant individuals shouldn’t include feelings of guilt/blame when there are many failures of systems that don’t provide enough support. It is indeed worrying, but the numbers and current state of affairs should just be reported as is.
- Para 3 Line 340 – This is more discussion like information which feels interesting and progressive – it still needs to be cut dramatically, can be repetitive but also jumps from idea to idea without clear connection but I would consider working more heavily on this section; providing some clear, concise ideas for how to improve PA in pregnancy in this context. Specific focus to how to target individuals <19 years, from semi-urban areas, low education, unemployed and nulliparous should be given as these were the groups identified as having lowest PA rates from the multivariate analysis.
- Remove WBOT acronym, just write out.
- Line 344 – ‘The advocacy philosophy imbibed by the UK guidelines with a dictum ‘every activity counts’ emphasising the need for inactive pregnant women to make an effort to accumulate PA’ – I heartily agree so hence my comments about the interpretations of these results – the individuals are still achieving a mean of 151 minutes of activity per week – it just isn’t exercise. I think some more discussion about this could be warranted – could women be better supported/encouraged if we reminding them that housework counts too rather than forcing exercise as an activity?
- Line 351 – ‘In this regard, healthcare providers (nurses, midwives, gynaecologists, physicians) dealing with pregnant women at the prenatal clinics should provide PA advice to pregnant women.’ I agree but is this feasible and realistic to ask of already over-burdened healthcare workers? Additionally, they likely receive no guidance or formal education on recommending physical activity, and may not have the expertise to modify or suggest activities beyond very broad statements that may actually be perceived as useless by pregnant individuals who want specific advice of what to do and what not to do. Should the onus be on healthcare providers to make these resources? Probably not – there needs to be multidisciplinary input including exercise physiologists, physiotherapists, and so on.
- Lines 367-377 - See comment 29. This paragraph and line 344 could be linked better.
- Lines 377-378 - I agree – so what exactly could be done? Could the talk test be explained as a marker for intensity for individuals? Could some examples be provided? Also the recommendation is only for moderate-intensity PA during pregnancy, not moderate-vigorous.
- Limitations - The PPAQ has been tested for validation across pregnant women in different countries/cultures – these papers should be referenced. Do you have any concerns about its use in a South African population given its development for North American populations? Although there are validation studies for the PPAQ, are there also publications showing limitations of this tool (in comparison to the IPAQ or objective Accelerometry data) that should be considered here for transparency?
Author Response
Reviewer #2
On behalf of my co-author, I would like to express appreciation for the thorough review and editorial work conducted on our manuscript. The comments provided by you are worthwhile; and has added significantly to the improvement of the manuscript. We have tried to address the queries point by point as instructed and made appropriate amendments as necessary.
Once again, many thanks.
General comments
The authors present findings from a quantitative assessment of prenatal physical activity behavior in a large cohort of pregnant individuals residing in the Eastern Cape, South Africa. The study does add to the scientific literature base, but really only to support that activity levels remain low in this population (as shown in nearly all pregnant populations around the world). The authors do complete a multivariate analysis to identify the demographic factors behind low activity levels in this region, and provide some ideas on interventions on how to improve physical activity during pregnancy. The manuscript is interesting but needs considerable revision to improve overall precision, clear rationale/explanations, and implications from this work.
Response
Thank you. Much appreciated.
At present, the introduction, results and discussion require significant editing for readability.
Response
These sections has been edited as per the comment above.
General comments:
- Consider avoiding acronym ‘PA.’ General consensus is that acronyms should be avoided where possible and if not established. PA won’t necessarily be recognised as ‘physical activity’ to researchers outside of this field.
Response
Thank you. The suggestion is noted, and corrections done.
- Instead of terming population as ‘pregnant women,’ authors should consider using more inclusive language in their manuscript (unless of course the authors asked all participants about their gender identity and they all responded ‘woman,’ if so, should be mentioned in the methods/results).
Response
The suggestion is noted, and corrections effected.
- Physical Activity Level during Pregnancyin South Africa: A Facility-Based Cross-Sectional Study
- Opening line of abstract: Physical activity (PA) participation during pregnancy confers many maternal and fetalhealth benefits.
Specific comments:
Abstract
- Consider avoiding acronym ‘PA.’ It may add to word count of abstract, but there are statements that could be removed to aid this change e.g. Line 13 – ‘as most studies have focused on the general population.’ Information not really needed.
Response
The acronym ‘PA’ has been corrected; and the statement ‘as studies have focused on the general population’ has been deleted as suggested.
- Line 20, ‘Statistical significance was set at p< 0.05 level.’ The word ‘level’ not needed.
Response
The word ‘level’ has been deleted as suggested.
Statistical significance was set at p< 0.05
- Line 21: what is the international guideline for prenatal activity? Do mean the general consensus of guidelines from around the world e.g. ACOG, CSEP, IOC, or do you mean the IOC? Either be specific as to which guideline or reword to something like ‘met recommendations’ for a more generic statement.
Response
It is the ACOG physical activity recommendation for during pregnancy. However, we have taken your suggestion to reflect a generic statement. Therefore, the sentence is reading thus:
Only 278 of the women (25.7%) met recommendations for prenatal activity (≥150 min moderate intensity exercise per week).
- Line 22, ‘the majority of women did not receive PA advice’ would be better following the statements regarding achievement of recommendations/actual activity levels (i.e. after sentence ending ‘…household activities..’
Response
The suggestion taken.
Most of the women participated in light exercises with a mean of 65.9%, and 63.7% participated in household activities. The majority of the women did not receive physical activity advice during prenatal care sessions (64.7%).
- Line 19 – acronym needed for adjusted odds ratio – AOR used later in abstract.
Response
“Factors negatively associated with prenatal physical activity were lower age (<19 years) (adjusted odd ratio (AOR)=0.3; CI: 0.16-0.76),...”
- Line 30 – low levels of PA during pregnancy in South Africa. Add to be specific, it is already pretty well established that PA is low in pregnancy, but this study is adding to that literature by focusing on a geographic location.
Response
Thank you for the suggestion.
The findings of this study demonstrated low levels of physical activity during pregnancy in South Africa
- Lines 31 to 34 is very repetitive of the results. I would consider condensing these together and using the final sentences to focus more on implications. I would like to see expansion on ideas of how to improve PA in this population.
Response
Various stakeholders concerns with maternal health should provide a general advocacy and workshops on the health benefits of leisure-time, and moderate-intensity physical activity during pregnancy. There need to provide pamphlets, display posters and other educational materials that contains information on prenatal activity; and use of text-messages on prenatal activity in pregnancy.
Introduction
- Lines 42 to 44 – you should also include references to the IOC recommendations, CSEP/SOGC, UK guidelines, etc. if you want to the guidelines around the world appropriately.
Response
The references for the various physical activity guidelines added and has been provided thus:
As such, specialised bodies and institutions such as the American College of Obstetricians and Gynaecologists (ACOG) [1], the U.S. Department of Health and Human Services (US DHHS) [2], World Health Organization (WHO) [3], the Joint Canadian Society for Exercise Physiology (CSEP)/Society of Obstetricians and Gynaecologists of Canada (SOGC) [4], Royal Australian and New Zealand College of Obstetrics and Gynaecology (RANZCOG) [5], International Olympic Committee (IOC) [6], Department of Health & Social Care, UK [7], Sports Medicine Australia [8], and the ACSM [9] recommend and encourage women, without contraindication, to engage in moderate-intensity exercise for 150 minute per week.
- Para 2 Lines 50 to 68 – This is all interesting and relevant information, but it could be considerably condensed. I think the authors just need to state reasons for PA participation are multifactorial - linked to demographics, perceived risk, and obstetric care advice, with some clear examples without extensive literature review.
Response
The paragraph has been revised as suggestion.
- Para 3 Lines 80 – 87 – Again, interesting information but not necessarily needed for the introduction – I think this could be moved to the discussion if not already included. It detracts from your rationale.
Response
The suggestion is taken. The information has been deleted and moved to the discussion section.
Methods
- Line 94 ‘Buffalo City…’ direct repetition of line 93. Remove.
Response
The repetition of Buffalo City Municipality in line 94 has been removed. The sentence has been revised to read:
The municipality is situated on the East Coast of the Eastern Cape Province.
- Line 125 – contraindications list. This is incorrect - these are not contraindications to prenatal exercise – these are reasons to cease activity if exercising. Contraindications include health conditions like preeclampsia, growth restriction, etc. Please check the reference again and amend. The ACOG, CSEP/SOGC and IOC have a clear list of contraindications to prenatal exercise. If you did not one of these lists to act as exclusion, then you must state that you did not exclude by contraindications, but reasons to cease exercise.
Response
The suggestion is taken. The women were excluded based on reasons to cease exercise. The reference has been provided thus:
Mottola, M.F.; Davenport, M.H.; Ruchat, S.; Davies, G.A.; Poitras, V.J.; Gray, C.E.; Jaramillo Garcia, A.; Barrowman, N.; Adamo, K.B.; Duggan, M.; et al. 2019 Canadian guideline for physical activity throughout pregnancy. Br. J. Sport Med. 2018, 40, 1339–1346.
- Figure 1 – what is the difference between incomplete PPAQ and missing data? I think you could benefit from stating what data was missing – was it antenatal outcome records? Also, if the PPAQ was interviewer-administered, how was it incomplete? I think this needs to be explained as your methods specifically states on Line 155-156 ‘To maximise the accuracy and ensure completeness of information, the PPAQ was interviewer-administered…’
Response
This has been corrected thus:
Of the 1215 pregnant women recruited, 42 participants did not meet the eligibility criteria, 26 declined to participate, and 65 had incomplete information on medical records data so were excluded. Finally, 1082 participants were included in the analysis.
- Line 175 - BMI underweight category is incorrect.
Response
Thank you for this keen observation. This was an oversight. This has been revised and also corrected thus.
We adopted the Institute of Medicine (IOM) recommended BMI cut-off values to classify underweight (18.5kg/m2), normal weight (18.5-24.9kg/m2), overweight (25.0-29.9kg.m2), and obese (>30.0 kg/m2) [45].
Results
- Line 207 to 209 – I think the statements about unplanned pregnancy and then PA advice, PA participation should be separated. They aren’t necessarily related and shouldn’t be in the same sentence.
Response
The sentence has been revised as suggested as follows:
Of the 1082 women, 731 (67.7%) affirmed their pregnancy was not planned. The majority of the women had not received PA advice (700; 64.7%). In addition, 704 (65.1%) did not participate in PA before pregnancy, and the majority of the women (753; 69.6%) never participated in PA in any of the trimesters.
- Table 1 – Would move headings to left-aligned. It is hard to read centralized with groupings beneath. This may be formatting edited later.
Response
The centralised formatting is from the journal.
- Table 4 and results section 3.3 – there is a lot of repetition for reporting of odds ratios in the text and in the table. It is not needed. The odds ratio should only be written in the text if they are not presented in the table or a figure. This is also applicable to the other results sections, but it is particularly noticeable here. It would make the results more succinct and far easier to read.
Response
The section has been revised considerably.
- Furthermore, more focus should be placed on the adjusted odds ratios than the crude odds ratios, at the moment the reporting of the crude odds ratios is overwhelming.
Response
As earlier stated above (Comment 19), the section has been revised.
- I am not quite understanding how 704 (65.1%) did not participate in PA during pregnancy, and the majority of the women (753; 69.6%) never participated in PA in any of the trimesters but the mean PA was 151 minutes per week (above recommendations!) and the median was 138.9. This doesn’t really fit with the ‘story.’ I think it is down to the wording. I would suggest being more precise with what PA is being referred too throughout the manuscript e.g. The majority of the population did not engage in moderate physical activity (i.e. sport/exercise score from PPAQ), but were physically active in light activities including household/caregiving, occupational and transportation activities. That is the distinction needed. The population are not meeting ‘exercise’ recommendations.
Response
Thank you for your observation and suggestion. The most activities participated in during pregnancy were light activities: household/caregiving, occupational and transportation activities. Light activities contributed to 65.9% to the total activity. In addition, the 704 (65.1%) refers to those who did not participate in physical activity before pregnancy. We have revised the section to read as follows:
The majority of the women did not engage in moderate physical activity (i.e. sport/exercise score from PPAQ), but were physically active in light activities including household/caregiving, occupational and transportation activities.
- On this note, more focus should be placed on encouraging pregnant individuals to increase leisure-time physical activity, and engaging in moderate activities.
Response
The suggestion is noted.
Discussion
- Specific focus on rephrasing to specify ‘moderate-intensity’ or ‘leisure time’ physical activity throughout.
Response
The suggestion is noted.
- The discussion is far too long and repetitive. There is a lot of information that provides insight but actually is not additive to the interpretation of these findings in this context. It feels more like an extended literature review than a discussion. Additionally, I often find the authors jump between different ideas within a paragraph with no clear link and this makes it hard to read. This section could be improved by condensing the ideas considerably and providing discussion of the most relevant works related to this study. Paragraphs 2, 4 and 5 need particular work.
Response
The discussion has been revised considerably as suggested.
- Line 307 – ‘This study used..’ comes out of nowhere, should be removed and addressed just in the limitations.
Response
This is deleted as suggested.
- There are a lot of emotive words e.g. worryingly, disappointingly – I think these should be removed. How physical activity messaging and engagement is shared with pregnant individuals shouldn’t include feelings of guilt/blame when there are many failures of systems that don’t provide enough support. It is indeed worrying, but the numbers and current state of affairs should just be reported as is.
Response
This is noted and deleted accordingly.
- Para 3 Line 340 – This is more discussion like information which feels interesting and progressive – it still needs to be cut dramatically, can be repetitive but also jumps from idea to idea without clear connection but I would consider working more heavily on this section; providing some clear, concise ideas for how to improve PA in pregnancy in this context. Specific focus to how to target individuals <19 years, from semi-urban areas, low education, unemployed and nulliparous should be given as these were the groups identified as having lowest PA rates from the multivariate analysis.
Response
Suggestion taken and the section revised.
- Remove WBOT acronym, just write out.
Response
WBOT acronym removed as suggested.
- Line 344 – ‘The advocacy philosophy imbibed by the UK guidelines with a dictum ‘every activity counts’ emphasising the need for inactive pregnant women to make an effort to accumulate PA’ – I heartily agree so hence my comments about the interpretations of these results – the individuals are still achieving a mean of 151 minutes of activity per week – it just isn’t exercise. I think some more discussion about this could be warranted – could women be better supported/encouraged if we reminding them that housework counts too rather than forcing exercise as an activity?
Response
The healthcare providers (nurses, midwives, gynaecologists, physicians) should encourage women to practice moderate physical activity during pregnancy; also emphasising to them, the importance of household physical activities in improving maternal health outcomes.
- Line 351 – ‘In this regard, healthcare providers (nurses, midwives, gynaecologists, physicians) dealing with pregnant women at the prenatal clinics should provide PA advice to pregnant women.’ I agree but is this feasible and realistic to ask of already over-burdened healthcare workers? Additionally, they likely receive no guidance or formal education on recommending physical activity, and may not have the expertise to modify or suggest activities beyond very broad statements that may actually be perceived as useless by pregnant individuals who want specific advice of what to do and what not to do. Should the onus be on healthcare providers to make these resources? Probably not – there needs to be multidisciplinary input including exercise physiologists, physiotherapists, and so on.
Response
In this regard, periodic workshops on prenatal physical activity, involving a multidisciplinary team (exercise physiologists, physiotherapists, gynaecologists, obstetricians) to provide talks on the importance of engaging in, and types of moderate-intensity activities for their health and the baby is advocated. Another possible intervention strategy is to involve other exercise specialists such as exercise physiologists, physiotherapists and biokineticists (practice exclusively in South Africa) to provide prenatal physical activity programmes, and social support within the community. By way of definition, biokineticists are exercise therapists that prescribe individualised exercise and physical activity for rehabilitation and promotion of health and quality of life [57]. In addition, utilising mHealth technology─MomConnect to promote awareness on prenatal physical activity and to encourage women to participate actively in household physical activities during pregnancy is a feasible option in South African context. The ‘MomConnect’ is a phone-based technological maternal health programme by the South African National Department of Health to support and encourage women to maintain and live healthy lifestyles during and after pregnancy, as well promote child health. The ‘MomConnect’ health promotion messages are freely available to the user. Research on exploring contextually relevant interventions to encourage pregnant women to increase leisure-time physical activity, and promote moderate-intensity activities during pregnancy in this setting is desirable.
- Lines 367-377 - See comment 29. This paragraph and line 344 could be linked better.
Response
There is need for context-specific interventions to educate and encourage pregnant women to participate in moderate-intensity physical activity to achieve optimal maternal health outcomes. The advocacy philosophy imbibed by the UK guidelines with a dictum ‘every activity counts’ emphasising the need for inactive pregnant women to make an effort to accumulate physical activity throughout the week [56], is something to learn from and apply in the South African context. In this regard, periodic workshops on prenatal physical activity, involving a multidisciplinary team (exercise physiologists, physiotherapists, gynaecologists, obstetricians) to provide talks on the importance of engaging in, and types of moderate-intensity activities for their health and the baby is advocated. Another possible intervention strategy is involving other exercise specialists such as exercise physiologists, physiotherapists and biokineticists (practice exclusively in South Africa) to provide prenatal physical activity programmes, and social support within the community. By way of definition, biokineticists are exercise therapists that prescribe individualised exercise and physical activity for rehabilitation and promotion of health and quality of life [57]. In addition, utilising mHealth technology─MomConnect to promote awareness on prenatal physical activity and to encourage women to participate actively in household physical activities during pregnancy is a feasible option in South African context. The ‘MomConnect’ is a phone-based technological maternal health programme by the South African National Department of Health to support and encourage women to maintain and live healthy lifestyles during and after pregnancy, as well promote child health. The ‘MomConnect’ health promotion messages are freely available to the user. Research on exploring contextually relevant interventions to encourage pregnant women to increase leisure-time physical activity, and promote moderate-intensity activities during pregnancy in this setting is desirable.
- Lines 377-378 - I agree – so what exactly could be done? Could the talk test be explained as a marker for intensity for individuals? Could some examples be provided? Also the recommendation is only for moderate-intensity PA during pregnancy, not moderate-vigorous.
Response
The section has been revised as follows:
There is need for context-specific interventions to educate and encourage pregnant women to participate in moderate-intensity physical activity to achieve optimal maternal health outcomes. The advocacy philosophy imbibed by the UK guidelines with a dictum ‘every activity counts’ emphasising the need for inactive pregnant women to make an effort to accumulate physical activity throughout the week [56], is something to learn from and apply in the South African context. In this regard, periodic workshops on prenatal physical activity, involving a multidisciplinary team (exercise physiologists, physiotherapists, gynaecologists, obstetricians) to provide talks on the importance of engaging in, and types of moderate-intensity activities for their health and the baby is advocated. Another possible intervention strategy is involving other exercise specialists such as exercise physiologists, physiotherapists and biokineticists (practice exclusively in South Africa) to provide prenatal physical activity programmes, and social support within the community. By way of definition, biokineticists are exercise therapists that prescribe individualised exercise and physical activity for rehabilitation and promotion of health and quality of life [57]. In addition, utilising mHealth technology─MomConnect to promote awareness on prenatal physical activity and to encourage women to participate actively in household physical activities during pregnancy is a feasible option in South African context. The ‘MomConnect’ is a phone-based technological maternal health programme by the South African National Department of Health to support and encourage women to maintain and live healthy lifestyles during and after pregnancy, as well promote child health. The ‘MomConnect’ health promotion messages are freely available to the user. Research on exploring contextually relevant interventions to encourage pregnant women to increase leisure-time physical activity, and promote moderate-intensity activities during pregnancy in this setting is desirable.
- Limitations - The PPAQ has been tested for validation across pregnant women in different countries/cultures – these papers should be referenced. Do you have any concerns about its use in a South African population given its development for North American populations? Although there are validation studies for the PPAQ, are there also publications showing limitations of this tool (in comparison to the IPAQ or objective Accelerometry data) that should be considered here for transparency?
Response
This section has been revised as suggested. We add as follows:
One obvious limitation of the study is the use of a self-reported questionnaire, which could have inherent bias in the manner of responses; and the PPAQ questionnaire was not validated locally against objective methods, as such, we cannot draw conclusion on its measurement properties (reliability, criterion validity, construct validity, responsiveness) in the South African context. Therefore, future studies on the validity of the PPAQ in this population is warranted; and studies using objective measure, such as accelerometer to evaluate physical activity during pregnancy. Furthermore, the PPAQ is reported to have insufficient construct validity in assessing total and vigorous physical activity scores in pregnancy, however, with low-to-moderate quality evidence [75]…. the prospective evaluation of physical activity using the PPAQ, a simple and short, reliable and valid tool, which has been used in several countries [44,76-81], and the large sample of pregnant women serve as unique strengths of the study.

Round 2
Reviewer 2 Report
My sincere thanks to the authors for making extensive changes in a short period. Overall, I think the manuscript is much improved, however I have a few additional comments/suggestions.
Comments:
- In my original review, I mentioned that more inclusive language should be considered throughout the manuscript. The authors made the two changes I suggested, but did not continue the edits throughout the manuscript i.e. the rest of the manuscript continues to read ‘pregnant women.’ I consider this to be up to editor preference, but I strongly recommend for future manuscripts (wherever submitted) that the authors use gender neutral language in their manuscript and consider adding gender identity questions to their methodology.
- Abstract - I still think line 23 – 25 in the abstract is unclear. The preceding sentences states that only 25.7% met recommendations (>150 mins per week) but the mean physical activity score was above that. I think it would be better to report the combined moderate-vigorous energy expenditure value here as this actually relates to the recommendations you are talking about e.g.
‘The average time spent in moderate-vigorous physical activity was ## minutes.’
This value would then be far less than 150 minutes and it would make far more sense with your overall conclusions of the manuscript. Also report the confidence levels within the abstract.
- Abstract – Line 32 specify intensity. The majority of women did not meet the recommendation of 150 minutes of moderate intensity activity per week.
- Abstract – Line 35 to 39 – My suggestion was to include applications of this work in the abstract, but I now consider these a bit too specific for the data that you are presenting – your results can’t speak to these interventions with certainty of efficacy. I would adjust to something more broad
‘In order to increase activity levels, future work should seek to improve knowledge, access and support for physical activity in pregnant individuals in South Africa. This should include education and advocacy regarding physical activity for professionals involved in maternal health provision.’
- Introduction – This is much improved. Thank you for making the revisions and edits.
- Introduction – Line 62 to 66, incredibly long sentence, would split the sentence. I also don’t know if the list of risks (e.g. miscarriage, growth restriction, etc.) is needed. It perpetuates misinformation. I would also be stronger in the initial part of the sentence ‘Despite no scientific evidence associated with adverse outcomes during or following prenatal physical activity…’ Meta-analyses from Davenport et al., BJSM 2018/2019 demonstrate this. I would suggest referencing these in addition to the reference from Coll. The first part of your statement needs more scientific backing than the latter.
- Methods – Line 117 – I would suggest that these factors are not ‘pre-existing health conditions.’ Please state this list as it is established – ‘Women with disabilities or reasons to cease exercise at the time of recruitment, such as…., were excluded.’
- Methods – Line 219 – Underweight category still incorrect, needs to be less than (e.g. <18.5kg/m2).
- Data Analysis – Line 224 to 226 – Please can you confirm the what metric was use to classify women as inactive or active. Was this total energy expended in minutes per week? Or the combined moderate-vigorous minutes per week? It should be the latter..
- Results – Line 262 – 266: These data are all available in the table, so does not need to be written out in this fashion. Also as with comment 2 above, I think your focus should be on the combined moderate-vigorous intensity activity, as this is what the recommendations are to meet.
- Results – Line 271 – 283: As above, direct repetition from the table. Summarize and direct to the table. These do not need to be written out.
- Results – Line 287 – 295: This has been improved, but I would still suggest that you do not need to write out each crude OR and confidence interval – they are available in the table.
- Discussion – Line 384: I struggle with ‘emphasising the household physical activities.’ Why household activities only?! This surely reinforces gender roles and I personally (and I am sure many females) would not be impressed if my healthcare provider emphasised this to me.. I would suggest editing to:
‘Healthcare providers (nurses, midwives, gynaecologists, physicians) should encourage women to practice moderate physical activity during pregnancy; also reinforcing the contribution of maintaining daily activities to maternal health.’
- Discussion – Line 402 – 406: Again, I think it is somewhat troublesome to suggest this is a choice – maybe it is that they don’t have access/education/support for other activities? I think this section (Lines 387 to 410) could be shortened and the general recommendations to be a little more forward thinking. I think Line 410 onwards is a better framework to focus on.
- Discussion Line 412 – new paragraph needed for ‘the advocacy..’
- Discussion Line 477-478 – reference or speculation? There are a few instances where there are speculative sentences included in the discussion. Although these may be true, they need to be referenced or removed. Other examples, Lines 489-490, Lines 397 to 400.
- Implications line 512 – consider ‘Healthcare providers should regard physical activity as a prescription, rather than an option’ for stronger take home.
Author Response
On behalf of my co-author, I would like to express appreciation for the comments/suggestions/ editorial work conducted on our manuscript. The comments from the Reviewer are worthwhile; and has further improved the manuscript. We have tried to address the Reviewer queries point by point as instructed and made appropriate amendments as necessary.
Once again, many thanks.
My sincere thanks to the authors for making extensive changes in a short period. Overall, I think the manuscript is much improved, however I have a few additional comments/suggestions.
Response
Thank you for the commendation. Much appreciated.
Comments:
- In my original review, I mentioned that more inclusive language should be considered throughout the manuscript. The authors made the two changes I suggested, but did not continue the edits throughout the manuscript i.e. the rest of the manuscript continues to read ‘pregnant women.’ I consider this to be up to editor preference, but I strongly recommend for future manuscripts (wherever submitted) that the authors use gender neutral language in their manuscript and consider adding gender identity questions to their methodology.
Response
- Abstract - I still think line 23 – 25 in the abstract is unclear. The preceding sentences states that only 25.7% met recommendations (>150 mins per week) but the mean physical activity score was above that. I think it would be better to report the combined moderate-vigorous energy expenditure value here as this actually relates to the recommendations you are talking about e.g.
‘The average time spent in moderate-vigorous physical activity was ## minutes.’
This value would then be far less than 150 minutes and it would make far more sense with your overall conclusions of the manuscript. Also report the confidence levels within the abstract.
Response
The average time spent in moderate-vigorous physical activity was 151.6 minutes (95% CI: 147.2-156.0). Most of the women participated in light exercises with a mean of 65.9% (95% CI: 64.8-67.0), and 47.6% (95% CI: 46.3-48.9) participated in household activities.
- Abstract – Line 32 specify intensity. The majority of women did not meet the recommendation of 150 minutes of moderate intensity activity per week.
Response
The sentence has been revised as suggested above.
The majority of women did not meet the recommendation of 150 minutes of moderate intensity activity per week.
- Abstract – Line 35 to 39 – My suggestion was to include applications of this work in the abstract, but I now consider these a bit too specific for the data that you are presenting – your results can’t speak to these interventions with certainty of efficacy. I would adjust to something more broad.
‘In order to increase activity levels, future work should seek to improve knowledge, access and support for physical activity in pregnant individuals in South Africa. This should include education and advocacy regarding physical activity for professionals involved in maternal health provision.’
Response
Thank you for the suggestion. We have revised this aspect as indicated above in the text.
- Introduction – This is much improved. Thank you for making the revisions and edits.
Response
Much appreciated.
- Introduction – Line 62 to 66, incredibly long sentence, would split the sentence. I also don’t know if the list of risks (e.g. miscarriage, growth restriction, etc.) is needed. It perpetuates misinformation. I would also be stronger in the initial part of the sentence ‘Despite no scientific evidence associated with adverse outcomes during or following prenatal physical activity…’ Meta-analyses from Davenport et al., BJSM 2018/2019 demonstrate this. I would suggest referencing these in addition to the reference from Coll. The first part of your statement needs more scientific backing than the latter.
Response
The sentence has been revised. The sentence is now reading thus:
Despite no scientific evidence associated with adverse outcomes during or following prenatal physical activity [25-28], seemingly, some pregnant women and obstetric care providers still hold doubts concerning the safety of prenatal physical activity.
In addition, we have added the following references, and accordingly, re-organise the reference numbering in text.
Davenport, M.H.; Ruchat, S.M.; Poitras, V.J.; Jaramillo Garcia, A.; Gray, C.E.; Barrowman, N.; Skow, R.J.; Meah, V.L; Riske L, Sobierajski, F.; James, M.; Kathol, A.J.; Nuspl, M.; Marchand, A.A.; Nagpal, T.S.; Slater, L.G.; Weeks, A.; Adamo, K.B.; Davies, G.A.; Barakat, R.; Mottola, M.F. Prenatal exercise for the prevention of gestational diabetes mellitus and hypertensive disorders of pregnancy: a systematic review and meta-analysis. Br. J. Sports Med. 2018, 52, 1367-1375.
Davenport, M.H.; Ruchat, S.M.; Sobierajski, F.; Poitras, V.J.; Gray, C.E.; Yoo, C.; Skow, R.J.; Jaramillo Garcia, A.; Barrowman, N.; Meah, V.L.; Nagpal, T.S.; Riske, L.; James, M.; Nuspl, M.; Weeks, A.; Marchand, A.A.; Slater, L.G.; Adamo, K.B.; Davies, G.A.; Barakat, R.; Mottola, M.F. Impact of prenatal exercise on maternal harms, labour and delivery outcomes: a systematic review and meta-analysis. Br. J. Sports Med. 2019, 53, 99-107.
Davenport, M.H.; Kathol, A.J.; Mottola, M.F.; Skow, R.J.; Meah, V.L.; Poitras, V.J.; Jaramillo Garcia, A.; Gray, C.E.; Barrowman, N.; Riske, L.; Sobierajski, F.; James, M.; Nagpal, T.; Marchand, A.A.; Slater, L.G.; Adamo, K.B.; Davies, G.A.; Barakat, R.; Ruchat, S.M. Prenatal exercise is not associated with fetal mortality: a systematic review and meta-analysis. Br. J. Sports Med. 2019, 53, 108-115.
- Methods – Line 117 – I would suggest that these factors are not ‘pre-existing health conditions.’ Please state this list as it is established – ‘Women with disabilities or reasons to cease exercise at the time of recruitment, such as…., were excluded.’
Response
The sentence has been revised as suggested thus:
Women with disabilities or reasons to cease exercise at the time of recruitment, such as ‘persistent excessive shortness of breath that does not resolve on rest, severe chest pain, regular and painful uterine contractions, vaginal bleeding, persistent loss of fluid from the vagina indicating rupture of the membranes, and persistent dizziness or faintness that does not resolve on rest’ [43], were excluded.
- Methods – Line 219 – Underweight category still incorrect, needs to be less than (e.g. <18.5kg/m2).
Response
This has been corrected thus:
We adopted the Institute of Medicine (IOM) recommended BMI cut-off values to classify underweight (<18.5kg/m2), normal weight (18.5-24.9kg/m2), overweight (25.0-29.9kg.m2), and obese (>30.0 kg/m2) [48].
- Data Analysis – Line 224 to 226 – Please can you confirm the what metric was use to classify women as inactive or active. Was this total energy expended in minutes per week? Or the combined moderate-vigorous minutes per week? It should be the latter.
Response
The women was classified as ‘active’ and ‘inactive’ based on their combined moderate-vigorous minutes per week.
- Results – Line 262 – 266: These data are all available in the table, so does not need to be written out in this fashion. Also as with comment 2 above, I think your focus should be on the combined moderate-vigorous intensity activity, as this is what the recommendations are to meet.
Response
The section revised as suggested thus:
The women’s physical activities were compared according to type and intensity of exercise (Table 2). Descriptive analysis of physical activity scores, as derived from the PPAQ scoring regarding the level of physical activity, showed the average time spent in moderate-vigorous physical activity was 151.6 minutes (95% CI: 147.2-156.0). The majority of the women did not engage in moderate-vigorous intensity physical activity (i.e. sport/exercise score from PPAQ), but were physically active in light activities including household/caregiving, occupational and transportation activities.
- Results – Line 271 – 283: As above, direct repetition from the table. Summarize and direct to the table. These do not need to be written out.
Response
Likewise, Table 3 presents the summary of the contribution of each physical activity level to the total score. The results show that, on average, light physical activity contributed most to the total activity. The participants rarely performed vigorous-intensity activities. Besides levels of activity, the contribution was also determined according to type of physical activity, namely, household, occupational, sport/exercise and transportation. Household activity contributed most to the total activity followed by occupational and transportation, while sport/exercise had the lowest contribution level. The rest is accounted for by physical inactivity.
- Results – Line 287 – 295: This has been improved, but I would still suggest that you do not need to write out each crude OR and confidence interval – they are available in the table.
Response
The crude ORs and confidence interval has been deleted in the text.
- Discussion – Line 384: I struggle with ‘emphasising the household physical activities.’ Why household activities only?! This surely reinforces gender roles and I personally (and I am sure many females) would not be impressed if my healthcare provider emphasised this to me. I would suggest editing to:
‘Healthcare providers (nurses, midwives, gynaecologists, physicians) should encourage women to practice moderate physical activity during pregnancy; also reinforcing the contribution of maintaining daily activities to maternal health.’
Response
Based on your suggestion, the sentence has been revised in the text as indicated above.
- Discussion – Line 402 – 406: Again, I think it is somewhat troublesome to suggest this is a choice – maybe it is that they don’t have access/education/support for other activities? I think this section (Lines 387 to 410) could be shortened and the general recommendations to be a little more forward thinking. I think Line 410 onwards is a better framework to focus on.
Response
This paragraph highlight the finding that majority of the women participate in household activities. We then make comparison with the literature, present the general picture physical activity participation in South African context regarding the types of physical activity, and provide plausible reasons for the participation of the women mostly in household activities. Some of these reasons have been alluded to by previous studies, which are referenced. We further added another possible reasons, based on your suggestion thus:
It might be possible that the women lack access, education and support for other physical activities, which explains why majority of the women are engaged in household activities.
- Discussion Line 412 – new paragraph needed for ‘the advocacy..’
Response
New paragraph created as suggested above.
- Discussion Line 477-478 – reference or speculation? There are a few instances where there are speculative sentences included in the discussion. Although these may be true, they need to be referenced or removed. Other examples, Lines 489-490, Lines 397 to 400.
Response
We deleted lines 477-478, 489-490, and 397-400 as suggested.
- Implications line 512 – consider ‘Healthcare providers should regard physical activity as a prescription, rather than an option’ for stronger take home.
Response
We appreciate your suggestion.
Healthcare providers should regard physical activity as a prescription, rather than an option.
